# First-Principles Calculations for Glycine Adsorption Dynamics and Surface-Enhanced Raman Spectroscopy on Diamond Surfaces

**DOI:** 10.3390/nano15070502

**Published:** 2025-03-27

**Authors:** Shiyang Sun, Chi Zhang, Peilun An, Pingping Xu, Wenxing Zhang, Yuan Ren, Xin Tan, Jinlong Yu

**Affiliations:** 1School of Mechanical Engineering, Inner Mongolia University of Science & Technology, Baotou 014010, Chinaxpp919@imust.edu.cn (P.X.); manifold@imust.edu.cn (W.Z.); renyuan_bt@126.com (Y.R.); heart_tan@126.com (X.T.); 2Beiben Trucks Group Co., Ltd., Baotou 014010, China

**Keywords:** diamond surface, glycine, SERS, AIMD, first-principles calculation

## Abstract

Based on first-principles calculations, the stability of three adsorption configurations of glycine on the (100) surface of diamonds was studied, leading to an investigation into the surface-enhanced Raman scattering (SERS) effect of the diamond substrate. The results showed that the carboxyl-terminated adsorption configuration (CAR) was the most stable and shortest interface distance compared to other configurations. This stability was primarily attributed to the formation of strong polar covalent bonds between the carboxyl O atoms and the surface C atoms of the (100) surface of diamonds. These results were further corroborated by first-principles molecular dynamics simulations. Within the temperature range of 300 to 500 K, the glycine molecules in the carboxyl-terminated adjacent-dimer phenyl-like (CAR) configuration exhibited only simple thermal vibrations with varying amplitudes. In contrast, the metastable ATO and carboxyl-terminated trans-dimer phenyl-like ring (CTR) configurations were observed to gradually transform into benzene-ring-like structures akin to the CAR configuration. After adsorption, the intensity of glycine’s characteristic peaks increased substantially, accompanied by a blue shift phenomenon. Notably, the characteristic peaks related to the carboxyl and amino groups exhibited the highest enhancement amplitude, exceeding 200 times, with an average enhancement amplitude exceeding 50 times. The diamond substrate, with its excellent adsorption properties and strong surface Raman spectroscopy characteristics, represents a highly promising candidate in the field of biomedicine.

## 1. Introduction

Most living cells derived from solid tissues require an adhering surface to live in vitro conditions [1]. The connection between the substrate and cells is based on the adsorption behavior of specific protein molecules [2]. Since proteins are primarily composed of folded amino acids, studying the interface between amino acids and inorganic material substrates is paramount for understanding how these cells adhere to surfaces outside the body. Glycine (Gly), a representative amino acid with a simple structure characterized by a carboxylic acid group, an amino group, and a CH_2_ linkage, lacks optical isomers [3]. Therefore, it serves as a logical starting point for gaining insights into the interplay between organic and inorganic substances. Previous extensive studies have investigated the adsorption behavior of glycine on various solid surfaces, such as alumina [4], iron [5], magnesium [6], and TiO_2_ [7]. The ideal conditions for substrate candidate materials include excellent biocompatibility, robust adaptability to various environments, and the ability to offer high detection sensitivity. Diamond, with its renowned biocompatibility and rich surface groups, offers versatile modifications and holds great promise for surface-enhanced Raman detection, thanks to its large scattering cross-section and wide bandgap.

The adsorption of glycine on diamond surfaces has garnered attention and research. Odbadrakh et al. [8] initially investigated the stable positioning of glycine-amino and carboxyl activities on the (001) surface of diamonds. Li et al. [9] further examined eight potential adsorption configurations of glycine-amino on the (100) surface of diamonds, and identified the benzene ring configuration as the most stable adsorption structure. The complexity of glycine’s adsorption structure arises from the diversity of its molecular structure [10], the change of carbon atoms in the diamond surface reconstruction process [11], and the interaction in three-dimensional space. Jinwoo [12] and Chatterjee et al. [13] observed five different adsorption positions of glycine on silicon surfaces through SEM experiments. This finding may indicate that the adsorption of glycine on inorganic substrates is not only affected by adsorption energy but also by environmental conditions. The accurate measurement of adsorption energy may not be a priority in the candidate process of biobased materials. Instead, it may be more critical to compare the adsorption capacity of different substrate materials and the impact of environmental conditions on them, as well as the excellent properties that candidate materials can provide during the detection process.

Among the various fields of biodetection technology, surface-enhanced Raman technology (SERS) has gradually emerged as a key direction for the detection of biological single molecules due to its unparalleled high sensitivity, accuracy, and specificity [14]. Diamond, as a SERS biological substrate, possesses abundant surface functional groups, a large scattering cross section, and wide band-gap regulation ability. In a comprehensive review by Szunerits [15], the purification and chemical reactivity of diamond surface functional groups were examined, highlighting the abundant and dense nature of these groups. Liu et al. [16] reported that nanodiamonds have been used as magnetic resonance imaging, photoacoustic imaging, and fluorescence imaging probes in clinical medicine and bioimaging fields. Zhang et al. [17] developed a boron-doped diamond substrate probe, and the SERS detection limit of methylene blue (MB) molecules could be as low as 10^−4^ M. It once again demonstrated the significant potential of diamond in SERS applications. However, certain molecules such as glycine often exhibit weak inherent Raman spectral signature signals [18] without the support of surface enhancement effects, thus making it difficult to be practically applicable. Although the infrared and absorption spectra of glycine on silicon substrates have been studied extensively [19], applying these results directly to diamond surfaces may lead to significant changes in the spectral properties of glycine. Therefore, whether glycine can achieve Raman enhancement on diamond surfaces and how this enhancement can be effectively achieved along with its underlying mechanism require further deep systematic discussion.

The dynamic evolution of glycine adsorption configurations on diamond surfaces was investigated using first-principles methods, and the variations in these configurations under different temperature conditions were analyzed. Furthermore, we examined the enhancement effect of the diamond substrate on the glycine Raman spectrum, particularly focusing on how interfacial atomic vibrations significantly amplify the Raman signal. This discovery holds significant potential for the application of diamond substrates in biodetection.

## 2. Computational Method and Details

Diamond substrates for biological interfaces are prepared using a variety of methods [20], including but not limited to detonation synthesis, chemical vapor deposition (CVD), high-pressure high-temperature (HPHT), and laser techniques. Regardless of the method used, the (100) surface is one of the most common exposed surfaces of diamonds [21]. The C atom on the ideal (100) surface has two suspended bonds and is therefore extremely chemically active [18]. In the case of vapor deposition, the carbon atoms of the (100) surface trap the H atoms, thus forming the H-terminal surface, and under vacuum and high pressure, the surface reconstruction of (100) will occur in the form of (1 × 1) [22] or (2 × 1) [22]. The H-terminal diamond exhibits chemical inertia, and its surface reactions typically involve the replacement of surface H atoms [22]. Conversely, the reactions occurring on the surface of nonterminated diamonds are intricately tied to their reconstructed structure. While these reactions are complex, they exert a pivotal influence on crucial surface chemical processes, including diamond growth and surface functionalization [23]. Consequently, the most widely accepted (2 × 1) reconstruction structure was adopted for this investigation, in which dangling bonds of adjacent surface carbon atoms tied together, forming a dimer with a π bond.

Glycine, the simplest natural amino acid, is characterized by the presence of carboxyl and amino groups at both ends of its carbon chain, as shown in Figure 1d. Upon adsorption onto a diamond’s surface, glycine exhibits three primary active sites, namely a nitrogen atom of an amino group and two oxygen atoms of a carboxyl group. During the adsorption process, in order to saturate the dangling bonds of glycine molecules and the diamond’s surface, the hydrogen atoms situated at these active sites tend to dissociate, and the C=O double bond undergoes cleavage within the glycine molecule while the π bonds open within the dimers present on the reconstructed diamond’s (001) surface [5]. Among the various adsorption structures, three adsorption structures were selected in the literature [4,5] with low energy, namely the open-loop structures of the amino-terminated phenyl-like ring, the carboxyl-terminated adjacent-dimer phenyl-like ring, and the carboxyl-terminated trans-dimer phenyl-like ring, recorded as ATO, CAR, and CTR, respectively, as shown in Figure 1. While previous studies had heavily relied on static energy calculations and the assessment of activation energies along predefined reaction paths, offering valuable insights into adsorption structures, it was crucial in the realm of biological interfaces to gain a deeper understanding of the environmental conditions and the evolution of various adsorption structures. Therefore, this study selected three typical adsorption structures and focused on observing their interfacial structures and bonding patterns, laying a foundation for analyzing the influence of temperature on these adsorption structures. The energy evaluation criterion of the adsorption structure is the adsorption energy value, which is expressed by Formula (1).*E_ad_* = *E_slab_* + *E_i_* − *E_tot_*
(1)
*E_ad_* represents the adsorption energy, *E_slab_* is the energy of the system before adsorption, *E_i_* is the energy of glycine, and *E_tot_* is the total energy after adsorption.

The adsorption structure was calculated using the VASP 6.4.1 software package [24]. The slab model was employed to establish the adsorption model. The diamond base comprised five layers, with the bottom two layers fixed and the top three layers relaxed. The fixed bottom surface was terminated with H atoms to mitigate the surface polarity. The glycine molecule is positioned on the relaxed surface above. To eliminate the effects of periodic boundary conditions, a substantial vacuum layer exceeding 20 Å was included above the model. Furthermore, a 4 × 4 lattice grid of diamond crystal was integrated into the slab surface, providing ample space for the migration of glycine molecules. In the computation, the electron correlations and exchange interactions were modeled precisely using the generalized gradient approximation (GGA) with the PBE exchange–correlation function [25]. A cutoff energy of 450 eV, accompanied by a Gaussian broadening factor of 0.05 eV, was implemented. Additionally, the Brillouin zone was meticulously sampled using a 3 × 3 × 1 k-point grid. The ion energies were converged to a threshold of 1 × 10^−5^ eV in order to achieve a stable structural configuration.

In delving into the intricate coordination relationship between glycine and the diamond’s reconstructed surface, mere reliance on a handful of adsorption structures determined solely by adsorption energy analysis proves inadequate, as the complexity of this interaction is difficult to capture in such a simplified manner. Even when we employ transition state theory to calculate the activation energies between various adsorption configurations, such calculations fail to accurately reflect the adsorption evolution of glycine molecules under actual temperature fields. To investigate the diffusion behavior of glycine molecules on the diamond’s surface and the effect of temperature on molecular activity, ab initio molecular dynamics (AIMD) simulations were applied. The simulations were performed using a Canonical NVT ensemble system with constant temperature control [26]. The environmental temperatures were set to 300 K, 400 K, and 500 K to simulate the room temperature, evaporative temperature [27], and decomposition temperature [28] for glycine molecules, respectively. Each simulation lasted for 10 ps, with a time interval of 1 fs for each step, resulting in a total of 10,000 steps executed to ensure sufficient sampling. The structural evolution of glycine was quantified using Root Mean Square Deviation (RMSD) [29], calculated according to Equation (2) as follows:(2)$$RMSD=\sqrt{\frac{1}{N}\sum_{i=1}^{N}\delta_{i}^{2}}$$
where *N* represents the number of atoms and $\delta_{i}$ represents the position offset.

Raman spectral information was performed on the Raman tensor obtained with polarizability calculations of vibrational modes of atoms under the two-harmonic approximation [30], as shown in Equation (3):(3)$$I^{Ram}=45({\frac{d\alpha }{dQ})}^{2}+7({\frac{d\beta }{dQ})}^{2}=45{\alpha \prime }^{2}+7{\beta \prime }^{2}$$
where(4)$$\alpha \prime =\frac{1}{3}({\tilde{\alpha^{\prime }}}_{xx}+{\tilde{\alpha^{\prime }}}_{yy}+{\tilde{\alpha \prime }}_{zz})$$(5)$${\beta \prime }^{2}=\frac{1}{2}\left[{\left({\tilde{\alpha \prime }}_{xx}{-\tilde{\alpha \prime }}_{yy}\right)}^{2}+{\left({\tilde{\alpha^{\prime }}}_{xx}-{\tilde{\alpha^{\prime }}}_{zz}\right)}^{2}{+\left({\tilde{\alpha^{\prime }}}_{yy}{-\tilde{\alpha^{\prime }}}_{zz}\right)}^{2}+6\left({\tilde{\alpha^{\prime }}}_{xy}^{2}+{\tilde{\alpha^{\prime }}}_{xz}^{2}+{\tilde{\alpha^{\prime }}}_{yz}^{2}\right)\right]$$
In the above equations, *α’* is the mean polarizability, *β’*^2^ is the anisotropy of the polarizability, *I^Ram^* is the Raman-scattering activity, and *Q* is the derivative of normal coordinates.

## 3. Results and Discussion

### 3.1. Molecular Adsorption Structure

Prior to investigating the glycine–diamond system, the glycine molecules were optimized geometrically, as shown in Figure 1. The outcomes indicated that the bond lengths and angle of the glycine molecule were consistent with the research of Ganji [31]. The intrinsic properties of the glycine molecule are predominantly governed by the amino and carboxyl groups within the molecular chain [32]. Alterations in the geometric configuration of these functional groups can markedly influence the overall performance of glycine.

As can be seen from Table 1, the adsorption energy (*E_ad_*) of CAR is the highest, 5.03 eV, indicating that this adsorption configuration is the most stable, consistent with the report in ref. [5]. The adsorption energies of ATO and CTR are similar, demonstrating a certain level of stability, albeit with differing underlying mechanisms. In the ATO configuration, an open-ring arm-like structure emerges from the interaction between the amino group and C atoms on the diamond’s surface, with minimal disruption to the amino acid’s molecular structure. This is evidenced by bond length fluctuations (*Δd_max_*) of merely 0.03 Å and bond angle adjustments (*Δθ_max_*) not exceeding 3.77°. This stability is primarily attributed to the formation of strong N–C bonds. In contrast, the CTR configuration involves the carboxyl group’s double oxygen atoms linking to diamond C atoms, requiring the conversion of C=O double bonds to C–O single bonds during adsorption, leading to substantial changes in the amino acid’s molecular structure, with *Δd_max_* of up to 0.08 Å and *Δθ_max_* of up to 9.82°. Consequently, the stability of the CTR configuration relies not only on the robust O–C bonds but also on the accommodation of these structural adjustments.

This phenomenon is primarily attributed to the significantly larger O–O distance in the carboxyl terminus compared to the C–C spacing on the reconstructed diamond surface. After the formation of C–O bonds at the interface, this spatial discrepancy prompts the carboxyl C atom within the glycine molecule to be “squeezed” into forming additional bonding with the diamond’s surface, resulting in a structure resembling a benzene ring, as illustrated in Figure 1b. This adsorption structure not only achieves a shorter adsorption distance (*l_avg_*) but also, according to Bader charge population calculations, facilitates a greater amount of charge transfer (*e_ct_*), thereby enhancing the adsorption stability.

According to the charge density difference map (CDD) analysis, the ATO configuration exhibits an obvious electron region along the N–C atomic chain at the interface, suggesting the formation of σ covalent bonds in this region. In Figure 1, red represents the electron-gained areas, while blue denotes the electron-depleted regions. Notably, a surrounding ring of electron-depleted area can be observed around the σ bonds, possibly due to the N–C formation of π bonds [33], as confirmed by the p-orbital hybridization of N–C atoms by a partial density of states (PDOS) analysis. Combined with the Bader charge population analysis based on the zero-flux surface, when the interface polarity is strongest, the C atom attached to the amino molecule on the diamond’s surface loses about 0.29 electrons. The CAR and CTR configurations exhibit more complex CDD patterns, featuring interfacial bonds primarily of σ and π types, and notably exhibit more pronounced polar covalent bonds compared to the ATO configuration. Additionally, in the CAR configuration, the C atom within the glycine molecule is also capable of forming distinct polar covalent bonds with the diamond’s surface.

As depicted in the PDOS (Figure 2), the N atom of the glycine molecule forms a bond with the diamond surface atom in the ATO adsorption configuration. In the energy range spanning from −9.8 eV to −3.7 eV, a prominent resonance phenomenon arises between the p-orbitals of the N atom and those of the neighboring C atom, peaking near −1.4 eV. This hybridization of p-orbital electrons signifies the existence of π bonding, confirming the circular electron-deficient region observed by CDD analysis. A similar phenomenon has been observed in the two different adsorption configurations of CAR and CTR. In the CAR configuration, the N–C bond exhibits a more pronounced effect, resulting in a resonance peak near −2.8 eV, which is mainly due to the difference in the electronic structure of the N–C bond and the O–C bond. In contrast, the CTR configuration exhibits a peak associated with the O–C bond within the positive energy range, indicating the emergence of the anti-π bond. This suggests that within the CTR configuration, the bonding strength is weaker than that observed in the CAR configuration, ultimately influencing the overall interfacial adsorption strength.

### 3.2. Temperature-Dependent Adsorption Behavior

Although AIMD may encounter some contingencies in simulating the behavior of molecules on inorganic interfaces, particularly during transition events, we conducted repeated calculation experiments up to 10 times on the same model and conducted a typical analysis of the phenomena that occurred repeatedly. Figure 3 illustrates the RMSD behavior of glycine molecules on the diamond’s surface under different adsorption configurations at 300 K. In the ATO adsorption configuration, the glycine structure undergoes two changes. Firstly, around 500 ps, the -CH_2_ group dehydrogenates and subsequently binds to a C atom on the diamond’s surface. Although this is significant, with the addition of C–C bonds to the interfacial bonding, significantly enhancing the adsorption stability, the overall configuration of the molecule does not alter substantially. Another change occurs around 3500 ps, where the COOH group rotates around its C atom, as depicted in the accompanying diagram. Similarly, in the CTR adsorption configuration, the glycine molecule undergoes multiple transformations, as illustrated in Figure 3. Initially, the carboxyl linkage across the diamond surface’s dimer breaks, and then the open-loop glycine molecule rotates around its main chain, with the amino terminal rotating prominently. Finally, glycine re-bonds on the diamond surface’s dimer, reaching a stable state basically similar to the CAR adsorption configuration. In the CAR adsorption configuration, the overall structure remains relatively stable, with the primary change centered on the potential rotation of the amino terminal of the molecule. However, this rotation is not an inevitable event, and its occurrence may be limited by a certain energy barrier, so it occurs with a certain probability. In addition, the molecule also experiences spring-like vibrations, primarily the wobbling of the H atom. It is evident that at room temperature, the CAR configuration of glycine adsorbed on the diamond’s surface is relatively stable, exhibiting only simple molecular thermal vibrations. Similarly, the ATO configuration also exhibits high stability, with simple adjustments to enhance C–C interfacial bonding sufficient to form stable adsorption. In contrast, the CTR configuration changes greatly, involving complex processes such as the breaking of interfacial bonds and molecular rotation, and can eventually be transformed into a more stable CAR configuration.

When discussing the effect of temperature on the adsorption behavior of glycine on the diamond’s surface, given that glycine molecules did not exhibit significant migration and diffusion behavior in AIMD simulation but were limited to thermal vibrations and local rotation around the molecular chain, the traditional mean squared displacement (MSD) analysis method, commonly used to study self-diffusion behavior, was not applicable in this context. Instead, RMSD was used to quantify the effect of temperature on the dynamics and stability of glycine molecule adsorption on the diamond’s surface, as shown in Figure 4. Taking the ATO structure as an example, the adsorption of glycine involves two primary processes: the sinking of the C atom and the rotation of the carboxyl O atom, as illustrated in Figure 4. The sinking of the methylene group leads to bonding with the C atom on the diamond’s surface, accompanied by a dehydrogenation reaction, forming a phenyl-like ring structure at the interface, as shown in the illustration. The rotation of the carboxyl group serves to align the main chain of the glycine molecule approximately on the same plane, thereby reducing the system’s energy. During this process, the carboxylate torsion angle of the glycine molecule is 168.6°. As a result, the final stable RMSD value reaches as high as 0.8 Å. The sinking of the glycine C atom to form a phenyl-like ring structure is a process that can be rapidly accomplished at all temperatures. However, the subsequent transition phase from this structure to the carboxyl rotation, as well as the vibration amplitude of the final structure in a thermal environment, both display variations with temperature changes. Despite the inherent randomness in the evolution of adsorption structures observed in AIMD simulations, it is undeniable that during the transition process, molecules have a higher probability of deviating from their stable positions, and upon reaching a stable state, the amplitude of molecular oscillations correspondingly increases. The temperature effects on the other two adsorption configurations are largely consistent with those observed in the ATO case. For the CAR configuration, as the temperature increases from 300 K to 500 K, there is only an increase in the molecular oscillation amplitude, while the basic morphology of the structure itself remains stable. As for the CTR structure, its evolution process is through interface bond breaking and molecular rotation, finally transforming into the CAR configuration. In this process, the increase in temperature primarily intensifies the rate of the transformation process and enhances the vibration amplitude of the final structure. Based on the simulation results of AIMD, it is confirmed that CAR configuration is the most stable adsorption structure of glycine on a diamond substrate. Meanwhile, the widely discussed amino-terminal ATO configuration undergoes structural transformation under normal and higher temperatures, forming a structure similar to a benzene ring. However, this transformation requires further dehydrogenation of the methylene group, and its adsorption energy is only 4.51 eV, weaker than that of the CAR configuration. Therefore, in the subsequent calculation of the enhancement effect of diamond substrate on glycine Raman spectra, the more stable CAR adsorption configuration was selected as the research basis.

### 3.3. Surface-Enhanced Raman Spectroscopy

In order to investigate the SERS effect of diamond substrates, it is necessary to compare the Raman spectra of glycine before and after adsorption. The calculation process of Raman spectra involves obtaining the vibrational information of atoms by calculating the polarizability and then deriving the Raman activity. Observed by glycine Raman spectroscopy, as shown in Figure 5, it is found that in the region below 2000 wavenumbers, the spectrum exhibits relatively weaker peaks. The main characteristic peaks were generally consistent with the literature [34], and the vibrational modes are presented in Table 2. The relatively strong peaks comprise the shear vibration of COO- at 760 cm^−1^, the stretching vibration of C–C at 815 cm^−1^, the shear vibration of CH_2_ and the rocking vibration of NH_3_ at 1439 cm^−1^, and the C=O vibrations at 1763 cm^−1^.

The Raman spectrum of the diamond’s (100) surface exhibits several distinct peaks in the low-frequency region. These include the C–C bending vibration peak (936 cm^−1^), stretching vibration peak (1368 cm^−1^), C=C vibration peak (1127 cm^−1^), and the SP3 bond shift vibration peak (1366 cm^−1^). Notably, these peaks differ significantly from the single characteristic peak observed in the diamond crystal (1332 cm^−1^). This variation arises from the predominant arrangement of surface atoms at the nanoscale, in contrast to the SP3 structure that primarily composes the crystal. Consequently, the differences in structure and atomic proportions lead to variations in vibration modes, thereby resulting in different characteristic peaks. Moreover, it is important to highlight that the peak around 1100 cm^−1^ exclusively appears in nanodiamonds and exhibits distinct vibration characteristics from the diamond crystal [35,36].

After glycine adsorbs on the diamond’s surface, the Raman scattering peaks retain the characteristic peaks of glycine and the diamond substrate, but the relative strengths changed obviously, and there was a significant blue shift (Figure 5). The characteristic peak at 1763 disappears due to the breakage of C=O bonds.

The glycine–diamond interface means that the COO- group of glycines was bonded to the surface’s C atoms, so the tensile (936 cm^−1^), bending (620 cm^−1^), and rocking (497 cm^−1^) vibration amplitudes associated with the COO- ion were enhanced. The vertical adsorption structure endowed NH_2_ ions with greater rotational freedom, coupled with the resonance effect of C–C bonds on the diamond’s surface, which enhanced the peak values of 1212 cm^−1^. At the same time, the disruption of the sp3 bonds of the carbon atoms on the diamond’s surface, caused by the adsorption structure, led to a noticeable reduction in the peak intensity observed at 1368 cm^−1^. The above theoretical calculation values are highly consistent with the experimental values [37,38].

The Raman peak of the glycine–diamond system exhibits a substantial increase compared to that of glycine alone and the diamond substrate. While the adsorption structure causes the characteristic peak of glycine to undergo a blueshift, a known phenomenon in SERS, the Raman spectra still indicate the presence of characteristic peaks of glycine molecules. The diamond substrate demonstrates a remarkably evident SERS effect, primarily attributed to the charge transfer of the COO group at the interface and the molecular resonance of NH_2_ groups. The highest enhancement amplitude exceeds 200 times, with an average enhancement amplitude exceeding 50 times.

The diamond substrate exhibits a significant enhancing effect on the Raman spectrum of glycine molecules, demonstrating notable selectivity. This characteristic proves advantageous for the detection of organic molecular features. Additionally, our findings suggest that the Surface-Enhanced Raman Spectroscopy (SERS) effect of the thin-film diamond substrate is correlated with its nanoscale properties. However, achieving more accurate Raman detection requires further in-depth research.

In summary, the diamond substrate material offers significant potential due to its exceptional adsorption capabilities and ability to enhance Raman signals. Consequently, it emerges as a promising alternative material for biointerfaces.

## 4. Conclusions

Based on the first-principles calculations, this study investigated the adsorption structure of glycine on a diamond substrate, discussed the influence of temperature on the surface migration activity of a glycine molecule, and analyzed the surface-enhanced Raman effect of a diamond substrate. The main findings are as follows:

Among the various adsorption configurations, the CAR structure exhibits the highest adsorption energy of about 5.03 eV—the smallest interfacial distance—mainly due to the formation of strong polar covalent bonds between carboxyl O atoms and the diamond surface’s C atoms, primarily including p–p-orbital hybridization. The result is further confirmed by the first-principles molecular dynamics simulations. Within the temperature range of 300 to 500 K, the glycine molecule in the CAR configuration exhibits only simple thermal vibrations of different magnitudes. In contrast, the other two adsorption configurations, ATO and CTR, exhibit metastable characteristics, undergo conformational evolution under the action of external energy (temperature), and eventually tend to form a benzene-like ring structure similar to CAR.

Although the Raman spectroscopic intensity of glycine molecules is relatively weak, upon adsorption onto the diamond’s surface, the intensity of its characteristic peaks significantly increases, accompanied by a blue shift of peak positions. Notably, the characteristic peaks related to carboxyl and amino groups exhibit marked enhancement, with the highest amplitude exceeding 200 times.

## Figures and Tables

**Figure 1 nanomaterials-15-00502-f001:**
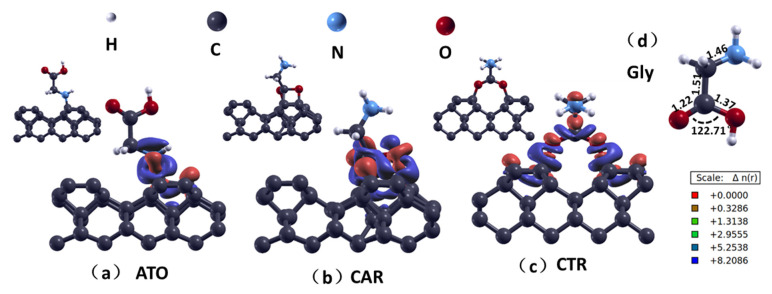
The structure of glycine adsorbed on the diamond’s (100) surface. The red region indicates electron gain, while the blue region indicates electron loss. (**a**) ATO configuration; (**b**) CAR configuration; (**c**) CTR configuration; (**d**) glycine molecular structure.

**Figure 2 nanomaterials-15-00502-f002:**
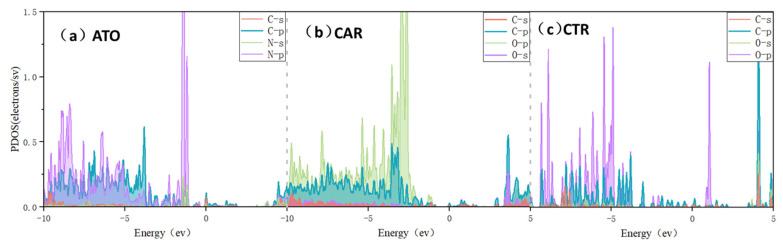
PDOS spectra of different adsorption structures of glycine on diamond’s surface. (**a**) ATO configuration; (**b**) CAR configuration; (**c**) CTR configuration.

**Figure 3 nanomaterials-15-00502-f003:**
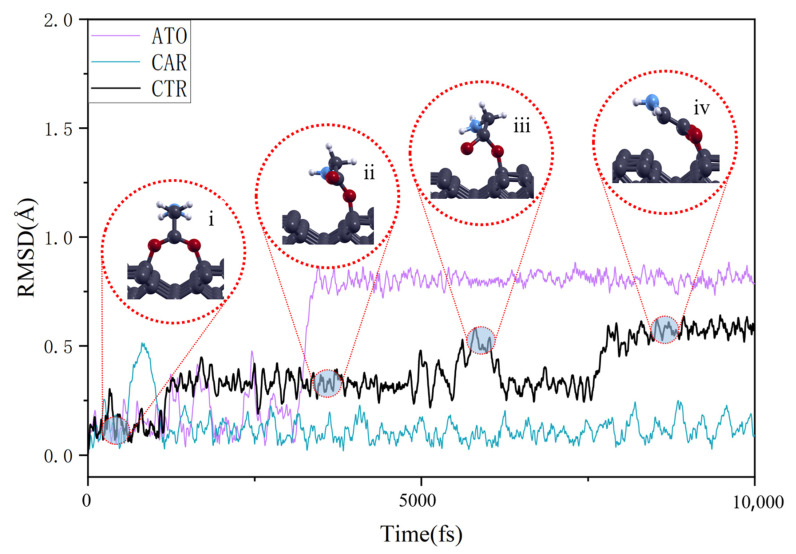
RMSD evolution curves of glycine adsorption on diamond substrate at 300 K. ATO, CAR, and CTR configurations are marked with purple, blue, and black lines, respectively.

**Figure 4 nanomaterials-15-00502-f004:**
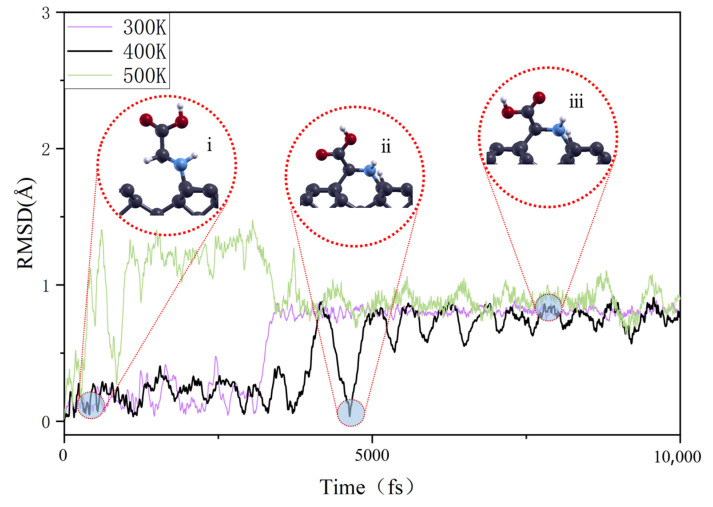
RMSD evolution curves of glycine adsorption on diamond substrate at various temperatures. 300 K, 400 K, and 500 K configurations are marked with purple, black, and green lines.

**Figure 5 nanomaterials-15-00502-f005:**
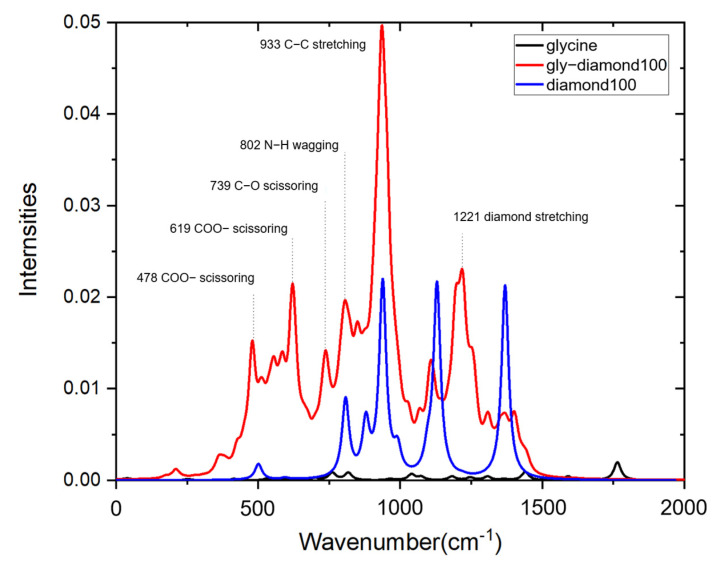
Raman spectra of glycine–diamond. Glycine, gly–diamond100, and diamond100 configurations are marked with black, red, and blue lines.

**Table 1 nanomaterials-15-00502-t001:** Various adsorption structures of glycine on diamond’s surface.

	*Δd_max_* (Å)	*Δθ_max_* (°)	*E_ad_* (eV)	*l_avg_* (Å)	*e_ct_* (e)
**ATO**	0.03	3.77	3.93	1.42	0.71
**CAR**	0.22	10.2	5.03	1.37	0.91
**CTR**	0.08	9.82	3.44	1.42	0.33

**Table 2 nanomaterials-15-00502-t002:** Raman spectra of glycine and glycine–diamond.

Wavenumber		Intensity		Vibration Mode
Glycine	Glycine–Diamond	Glycine	Glycine–Diamond	
251	210	1.51 × 10^−4^	1.2 × 10^−3^	C–O bridge connection symmetric stretching
411	368	1.76 × 10^−4^	2.8 × 10^−3^	C–N scissor bending
532	478	2.5 × 10^−4^	1.5 × 10^−2^	C–O scissor bending
698	619	3.2 × 10^−4^	2.1 × 10^−2^	C–O scissor bending
760	734	9.08 × 10^−4^	1.4 × 10^−2^	C–O scissor bending
815	806	8.85 × 10^−4^	0.02	N–H out of plane wagging
961	935	1.94 × 10^−4^	0.05	C–C symmetric stretching
1038	1023	7.33 × 10^−4^	8.5 × 10^−3^	C–N symmetric stretching
1072	1069	4.9 × 10^−4^	8 × 10^−3^	C–C–N asymmetric stretching
1181	1107	4.73 × 10^−4^	1.3 × 10^−2^	C–H out of plane twisting
1306	1365	4.64 × 10^−4^	7.4 × 10^−3^	C–C scissor bending
1439	1398	9.41 × 10^−4^	7.5 × 10^−3^	C–H scissor bending

## Data Availability

The data that support the findings of this study are available on request from the corresponding author. The data are not publicly available due to state restrictions such as privacy or ethical restrictions.

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
