# Peer review of "First-Principles Calculations for Glycine Adsorption Dynamics and Surface-Enhanced Raman Spectroscopy on Diamond Surfaces"

_nanomaterials, 2025, doi:10.3390/nano15070502_

Round 1
Reviewer 1 Report
Comments and Suggestions for Authors
The authors should consider the following points:
The study relies entirely on computational modeling without experimental confirmation of the Raman enhancement effect.
A discussion on how the computational findings compare to real-world SERS experiments on diamond surfaces is recommended
If experimental work is not possible, suggest the authors cite previous experimental studies that validate similar computational predictions.
The study focuses on glycine adsorption on diamond, however it can also be beneficial to compare it with other functionalized surfaces (e.g., oxide coatings, metal-doped diamond, or hybrid nanomaterials).
The authors should briefly discuss practical challenges in fabricating diamond-based SERS substrates and how these findings could be applied in real biosensor devices.
While the computational study is well-structured, real-world applications require considerations of scalability and material fabrication.
Author Response
Dear Reviewer,
We are very grateful to your critical comments and thoughtful suggestions. Based on these comments and suggestions, we have made careful modification on the original manuscript. All changes made to the text are in red in the revised manuscript so that they may be easily identified. Some of your questions were answered below.
The principle of Raman spectral calculation is based on Density Functional Theory (DFT) and Density Functional Perturbation Theory (DFPT). According to DFPT, the dynamical matrix of the system is obtained through second-order perturbation calculations of the system's Hamiltonian. The dynamical matrix, which correlates with atomic vibrational frequencies, has eigenvalues and eigenvectors corresponding to vibrational frequencies and modes respectively. In VASP implementations, the dynamical matrix is derived by calculating the second derivative of the system's energy through applying minute displacement perturbations. Diagonalization of this matrix yields eigenvalues and eigenvectors, where the square roots of eigenvalues represent the frequencies of Raman-active vibrational modes, and the eigenvectors describe the displacement directions and magnitudes of atoms in corresponding vibrational modes. Current computational spectroscopy techniques have achieved sufficient maturity, demonstrating good consistency with experimental values [1].
In this study, the calculated Raman spectra of diamond and glycine show excellent agreement with experimental data [2,3] in peak positions. For instance: The experimental value for [COO]⁻ vibration is 606 cm⁻¹ versus the calculated 619 cm⁻¹.Diamond's Raman peak appears at 1131 cm⁻¹ experimentally versus 1125 cm⁻¹ computationally.
The close alignment between computational and experimental results validates the accuracy of Raman spectral simulations. This theoretical verification forms the foundation for the current research endeavor.
[1] Bagheri M, Komsa H P. High-throughput computation of Raman spectra from first principles[J]. Scientific Data, 2023, 10(1): 80.
[2] Prawer S, Nemanich R J. Raman spectroscopy of diamond and doped diamond[J]. Philosophical Transactions of the Royal Society of London. Series A: Mathematical, Physical and Engineering Sciences, 2004, 362(1824): 2537-2565.
[3] Parameswari A, Premkumar S, Premkumar R, et al. Surface enhanced Raman spectroscopy and quantum chemical studies on glycine single crystal[J]. Journal of Molecular Structure, 2016, 1116: 180-187.
Thank you again for your expertise. Please do not hesitate to contact us if additional revisions are needed.
Kind regards
Sincerely yours
ShiYang Sun
Reviewer 2 Report
Comments and Suggestions for Authors
- Line 12: could the respected authors explain why the study justifies the choice of glycine as a model amino acid for adsorption studies and no other amino acids
- Line 23: could the authors explain why the diamond substrate could enhance Raman signals compared to other substrates?
- Line 23-24: Please add some findings values of the Raman data
- Line 43: Please indicate in detail the key advantages of using diamond as a substrate for SERS applications.
- Line 91: indicate if there are any alternative methods that could be used for preparing diamond substrates that might influence adsorption behavior.
- Line 127: How does the formula for adsorption energy compare with previous studies on similar systems?
- Line 152: Please demonstrate how the AIMD parameters could impact the accuracy of the adsorption behavior predictions.
- Line 198: Please explain the importance of the benzene ring-like structure formed in the adsorption process.
- Line 231: Could the respected authors explain how the PDOS spectra support the findings related to the stability of different adsorption configurations?
- Line 345: Please indicate how reliable are the Raman spectra calculations in predicting experimental SERS effects and if there are any other methods to confirm the study findings.
Author Response
Dear Reviewer,
Thank you sincerely for taking the time to review our manuscript and providing valuable feedback. Your insightful comments have greatly contributed to improving the quality of this work. All changes made to the text are in red in the revised manuscript so that they may be easily identified. We have carefully addressed each of your concerns, and our responses are outlined below:
- Line 12: could the respected authors explain why the study justifies the choice of glycine as a model amino acid for adsorption studies and no other amino acids.
The molecular structure of glycine exhibits simplicity, while its dual functional groups (carboxyl and amino groups) enable synergistic investigation of interfacial adsorption through multiple mechanisms. Therefore, glycine was selected as the adsorbate in this study.
- Line 23: could the authors explain why the diamond substrate could enhance Raman signals compared to other substrates?
First, the abundant functional groups on diamond surfaces enable directional functionalization through chemical bonding, allowing precise modulation of the coupling between surface plasmon resonance effects and molecular recognition capabilities. Second, the exceptionally high scattering cross-section significantly enhances Raman signal amplification factors. Compared to traditional substrates: Metallic substrates may interfere with or damage bioactive substances. Semiconductor substrates exhibit lower scattering efficiency, typically requiring surface roughness or nanostructuring for indirect signal enhancement.
In contrast, diamond's intrinsic optical properties facilitate more efficient signal amplification while maintaining biocompatibility.
- Line 23-24: Please add some findings values of the Raman data
We were really sorry for our careless mistakes. Thank you for your reminder.
In response, we have changed: After adsorption, the intensity of glycine's characteristic peaks increased substantially, accompanied by a blue shift phenomenon. Notably, the characteristic peaks related to the carboxyl and amino groups exhibited the highest enhancement amplitude exceed-ing 200 times, with an average enhancement amplitude exceeding 50 times.
- Line 43: Please indicate in detail the key advantages of using diamond as a substrate for SERS applications.
Diamond demonstrates unique advantages in surface-enhanced Raman spectroscopy (SERS) research, primarily stemming from the synergistic interaction between its physicochemical properties and functionalization potential. First, its exceptional biocompatibility provides an ideal substrate for non-destructive detection of biomolecules. Second, the abundant functional groups on diamond surfaces enable directional functionalization through chemical bonding, allowing precise modulation of the coupling between surface plasmon resonance effects and molecular recognition capabilities. Additionally, its wide bandgap effectively suppresses background noise induced by photoinduced electron transfer, while the high scattering cross-section significantly enhances Raman signal amplification factors.
- Line 91: indicate if there are any alternative methods that could be used for preparing diamond substrates that might influence adsorption behavior.
Common methods for nanodiamond synthesis include chemical vapor deposition (CVD) and detonation synthesis. Detonation method utilizes the extreme temperature/pressure from explosive detonation to force direct phase transition of carbon precursors (e.g., graphite) into diamond. However, the resultant material contains unstable sp²/sp³ hybrid phases, metallic residues, lattice distortions, and randomly distributed surface functional groups, making it unsuitable for precision devices like SERS substrates.
In contrast, CVD grows diamond through gas-phase carbon precursors (e.g., methane) that decompose under controlled energy excitation, enabling layer-by-layer carbon deposition on substrates. This approach yields high-crystallinity products with minimal defects, particularly suitable for fabricating high-purity functionalized nanodiamonds for precision instrumentation.
Key advantages of CVD: Tunable process parameters (gas composition: CH₄/H₂/O₂; plasma conditions) enable direct generation of uniform active groups (-OH, -COOH) on diamond surfaces. Controlled functional group distribution facilitates systematic investigation of adsorption behaviors
- Line 127: How does the formula for adsorption energy compare with previous studies on similar systems?
Adsorption energy, defined as the key parameter quantifying the stability of molecular or atomic adsorption processes on material surfaces, exhibits significant variations in calculation formulae and sign conventions depending on research systems and computational methodologies [1]. While different literature sources adopt distinct mathematical formulations, the fundamental trends remain consistent, with discrepancies primarily arising from differences in sign conventions.
[1] Li W X, Stampfl C, Scheffler M. Oxygen adsorption on Ag (111): A density-functional theory investigation[J]. Physical Review B, 2002, 65(7): 075407.
- Line 152: Please demonstrate how the AIMD parameters could impact the accuracy of the adsorption behavior predictions.
In Ab Initio Molecular Dynamics (AIMD) simulations, three core parameters govern physical realism: temperature, time, and ensemble [1]. Temperature is regulated via thermostat coupling to maintain thermodynamic equilibrium by controlling atomic kinetic energy distributions. Time parameters (time step and total simulation duration) dictate the accuracy and observability of dynamical evolution. Ensemble constraints (e.g., NVT, NPT) fix macroscopic system states (energy, volume, etc.) under predefined thermodynamic conditions. The synergistic interplay of these parameters collectively ensures both the physical authenticity of simulated processes and the statistical reliability of derived results.
[1] Zhang H, Shang S L, Wang W Y, et al. Structure and energetics of Ni from ab initio molecular dynamics calculations[J]. Computational materials science, 2014, 89: 242-246.
- Line 198: Please explain the importance of the benzene ring-like structure formed in the adsorption process.
The planar rigidity of the benzene ring structure shortens the adsorption distance and enhances stability, while simultaneously facilitating increased charge transfer.
- Line 231: Could the respected authors explain how the PDOS spectra support the findings related to the stability of different adsorption configurations?
Projected Density of States (PDOS) elucidates electronic interactions (e.g., orbital hybridization, charge transfer) between adsorbates and substrates by decomposing contributions from specific atoms/orbitals to electronic states. These interactions directly govern the stability of adsorption configurations. PDOS analysis enables systematic investigation of bonding characteristics, providing insights into bond formation mechanisms and relative strengths.
Case study: In the ATO adsorption configuration, PDOS reveals a pronounced resonance between the p-orbitals of N atoms and adjacent C atoms within the energy range of -9.8 eV to -3.7 eV, peaking near -1.4 eV. This p-orbital hybridization indicates π-bonding interactions. Similar phenomena are observed in both CAR and CTR adsorption configurations.
- Line 345: Please indicate how reliable are the Raman spectra calculations in predicting experimental SERS effects and if there are any other methods to confirm the study findings.
The principle of Raman spectral calculation is based on Density Functional Theory (DFT) and Density Functional Perturbation Theory (DFPT). According to DFPT, the dynamical matrix of the system is obtained through second-order perturbation calculations of the system's Hamiltonian. The dynamical matrix, which correlates with atomic vibrational frequencies, has eigenvalues and eigenvectors corresponding to vibrational frequencies and modes respectively. In VASP implementations, the dynamical matrix is derived by calculating the second derivative of the system's energy through applying minute displacement perturbations. Diagonalization of this matrix yields eigenvalues and eigenvectors, where the square roots of eigenvalues represent the frequencies of Raman-active vibrational modes, and the eigenvectors describe the displacement directions and magnitudes of atoms in corresponding vibrational modes. Current computational spectroscopy techniques have achieved sufficient maturity, demonstrating good consistency with experimental values [1].
In this study, the calculated Raman spectra of diamond and glycine show excellent agreement with experimental data [2,3] in peak positions. For instance: The experimental value for [COO]⁻ vibration is 606 cm⁻¹ versus the calculated 619 cm⁻¹.Diamond's Raman peak appears at 1131 cm⁻¹ experimentally versus 1125 cm⁻¹ computationally.
The close alignment between computational and experimental results validates the accuracy of Raman spectral simulations. This theoretical verification forms the foundation for the current research endeavor.
[1] Bagheri M, Komsa H P. High-throughput computation of Raman spectra from first principles[J]. Scientific Data, 2023, 10(1): 80.
[2] Prawer S, Nemanich R J. Raman spectroscopy of diamond and doped diamond[J]. Philosophical Transactions of the Royal Society of London. Series A: Mathematical, Physical and Engineering Sciences, 2004, 362(1824): 2537-2565.
[3] Parameswari A, Premkumar S, Premkumar R, et al. Surface enhanced Raman spectroscopy and quantum chemical studies on glycine single crystal[J]. Journal of Molecular Structure, 2016, 1116: 180-187.
Once again, we acknowledge your comments and constructive suggestions very much, which are valuable in improving the quality of our manuscript.
Best regards
ShiYang Sun
Round 2
Reviewer 1 Report
Comments and Suggestions for Authors
The authors have addressed the comments.
Reviewer 2 Report
Comments and Suggestions for Authors
I greatly appreciate that the authors have thoroughly revised, restructured, and completed the manuscript as suggested. After the revision, the manuscript is more straightforward, and its scientific quality has improved significantly. I agree with the modifications made by the authors and believe the paper can be accepted in its current form.